# Research Hotspots and Trends of Bone Xenograft in Clinical Procedures: A Bibliometric and Visual Analysis of the Past Decade

**DOI:** 10.3390/bioengineering10080929

**Published:** 2023-08-04

**Authors:** Jiayue Li, Yujue Zhao, Shili Chen, Simin Wang, Wen Zhong, Qing Zhang

**Affiliations:** 1Guangzhou Key Laboratory of Basic and Applied Research of Oral Regenerative Medicine, Guangdong Engineering Research Center of Oral Restoration and Reconstruction, Affiliated Stomatology Hospital of Guangzhou Medical University, Guangzhou 510182, China; 2Laboratory for Myology, Department of Human Movement Sciences, Faculty of Behavioural and Movement Sciences, Vrije Universiteit Amsterdam, Amsterdam Movement Sciences, 51081 BT Amsterdam, The Netherlands

**Keywords:** bibliometric analysis, CiteSpace, bone defects, bone xenograft, research hotspots

## Abstract

Background: Bone defect therapy is a common clinical challenge for orthopedic and clinical physicians worldwide, and the therapeutic effect affects the physiological function and healthy life quality of millions of patients. Compared with traditional autogenous bone transplants, bone xenografts are attracting attention due to their advantages of unlimited availability and avoidance of secondary damage. However, there is currently a lack of bibliometric analysis on bone xenograft. This study aimed to use bibliometric methods to analyze the literature on bone xenograft from 2013 to 2023, to explore the current status, hotspots, and future trends of research in this field, and to promote its development and progress. Methods: Using the Web of Science Core Collection database, we retrieved and collected publication data related to xenogeneic bone grafting materials worldwide from January 2013 to March 2023. Origin (2021), CiteSpace (6.2.R2 standard), and an online bibliometric platform were used for bibliometric analysis and data visualization. Results: A total of 3395 documents were retrieved, and 686 eligible papers were selected. The country and institutions with the highest number of publications and centrality were the United States (125 papers, centrality = 0.44) and the University of Zurich (29 papers, centrality = 0.28), respectively. The most cited author was Araujo MG (163 times), and the author with the most significant centrality was Froum SJ (centrality = 0.09). The main keyword clusters were “tissue engineering”, “sinus floor elevation”, “dental implants”, “tooth extraction”, and “bone substitutes”. The most significant bursting keywords in the last three years were “platelet rich fibrin”. Conclusions: Research on bone xenograft is steadily growing and will continue to rise. Currently, research hotspots and directions are mainly focused on dental implants related to bone-augmentation techniques and bone tissue engineering. In the future, research hotspots and directions may focus on decellularization technology and investigations involving platelet-rich fibrin.

## 1. Introduction

Bone defects represent a common clinical challenge for orthopedic and clinical physicians worldwide [1]. Millions of patients suffer from bone defects each year due to trauma, infection, tumor resection, or congenital malformations, with the global burden of these diseases continuing to rise [2]. Successful treatment of bone defects is crucial for restoring the structural integrity, function, and overall quality of life of the affected patients [3,4]. Bone regeneration is an important method and research direction for treating bone defects [5,6,7]. Autologous bone graft has traditionally been the gold standard for bone regeneration, but they are not always feasible options due to limited donors, prolonged surgical time, and potential donor site morbidity [8]. Therefore, alternative bone graft materials such as bone xenografts have been explored as potential substitutes [9]. In recent years, bone xenografts have become a viable alternative to autografts for bone grafts due to their availability, cost-effectiveness, and reduced morbidity at donor sites [10]. However, despite the increasing use of bone xenografts, it is still necessary to understand the current research status of xenografts, including the hotspots and trends.

Bibliometrics is a quantitative method for analyzing scientific literature and has been proven to be an indispensable tool for tracking research trend evolution and identifying high-impact nodes in a given field [11]. Therefore, this bibliometric study aims to reveal the hotspots and trends in xenograft research to provide valuable insights for researchers, clinical physicians, and policymakers [12]. By studying the advances in this rapidly developing life science field, we can better understand the current state of the art and identify opportunities for future innovations.

In this bibliometric study, we will analyze scientific publications related to bone xenograft over the past decade. Our analysis will focus on exploring publication trends and highlighting contributing countries, institutions, and authors. We will also use various statistical tools to identify the most relevant research topics and keywords related to bone xenograft. In addition, we will recognize prominent research themes, collaborations, and potential directions for future research. Overall, the results of this bibliometric study can provide important insights into the current research status of bone xenograft, including the hotspots and trends (Figure 1). These insights can help improve and develop new bone xenografts for clinical use and ultimately improve patient outcomes.

## 2. Materials and Methods

### 2.1. Data Collection

The first author conducted a literature search on the platform of Web of Science (WoS) in the “Web of Science Core Collection (WoSCC)” database using the search formula “((ALL = (xenogeneic bone)) OR ALL = (bone xenograft)) OR ALL = (bone xenotransplantation)” on 28 March 2023. The search period was from 2013 to 2023, and there were no restrictions on article types. A total of 3395 articles were retrieved. Two researchers conducted independent screening based on inclusion and exclusion criteria, and after review and verification, 686 articles were selected. The inclusion criteria were: (1) the article type was a research paper or review paper. (2) The research content was related to “research and application of bone xenografts of bone defects”. The exclusion criteria were: (1) Meeting abstract, proceeding paper, letter, news item, editorial materials, book chapter, and retracted publications. (2) Duplicates. (3) Articles unrelated to bone xenografts, such as organ transplantation, bone marrow transplantation, cancer bone metastasis, cardiovascular diseases, and cartilage tissue engineering. The screened literature was downloaded in the format of “fully documented and cited references”, including information such as title, abstract, keywords, publication year, authors, nationality, journal name, research direction, publishing institution, funding agencies, and references. That was saved as a plain text file in the raw analysis data sample (Figure 2).

### 2.2. Bibliometric and Visualization Analysis

Information on the number of annual publications, research types, and other relevant data was extracted from the sample data. The data analysis software Origin (2021) was used to conduct a descriptive analysis of the data and create bar graphs. Additionally, we conducted visualization analysis using the CiteSpace (6.2.R2 Standard) software and the bibliometric online platform https://bibliometric.com/, accessed on 31 March 2023. With CiteSpace, we explored the collaboration or co-citation network of countries, institutions, authors, references, and keywords while identifying references and keywords clusters and burst items. On the https://bibliometric.com/ online platform, we conducted country cooperation relationship analysis to obtain statistical data and visualize knowledge maps.

The knowledge map produced by CiteSpace provides research information conducted over a specific time range, with node or edge colors referencing the color bar indicating the year. Node size represents the frequency of occurrence, with larger nodes indicating higher frequency. The circular layers of the node represent the annual rings, with a purple circular layer marking the centrality (when the number of centralities is larger than 0.1) and its width indicating the magnitude of the centrality, reflecting the structural and influential aspects of countries, institutions, authors, keywords, and references.

By analyzing the knowledge map and relevant literature, we interpret the current research status of bone xenografts and analyze their hotspots and trends.

## 3. Results

### 3.1. Publication of Annual Trend

Annual publication trends were analyzed by counting the number of papers published each year using documents obtained from the WoSCC. To reflect changes in the volume of the literature published over the past decade, a column chart and cumulative publication line graph were plotted (Figure 3). The chart shows a steady upward trend in the number of publications per year. According to the analysis of the WoS platform, the H-index of bone xenograft has been 41 over the past decade.

### 3.2. Analysis of National (Regional) Collaboration

To identify the countries and regions with a significant number of publications and influential contributions of bone xenografts, we conducted a country (regional) cooperation analysis of the relevant literature. The national cooperation network diagram indicates a total of 63 nodes, 245 connections, and a network density of 0.1254 (Figure 4). Figure 5 presents the publication volume and centrality of the top 10 countries in terms of publication output. The United States is the most productive and influential country of international cooperation on bone xenograft, with 125 papers and a centrality of 0.44. Following the United States, the top five countries include Italy (123 papers, 0.16 centrality), Brazil (75 papers, 0.11 centrality), China (74 papers, 0.10 centrality), Germany (74 papers, 0.33 centrality), and South Korea (64 papers, 0.01 centrality). Moreover, countries with significant centrality (>0.1) include the United States, Germany, Switzerland (63 papers, 0.18 centrality), Italy, Brazil, Spain (62 papers, 0.11 centrality), France (15 papers, 0.11 centrality), and China. Notably, two highly cited papers published by France are “High-Temperature Sintering of Xenogeneic Bone Substitutes Leads to Increased Multinucleated Giant Cell Formation: In Vivo and Preliminary Clinical Results” [13] and “MicroRNA 210 Mediates VEGF Upregulation in Human Periodontal Ligament Stem Cells Cultured on 3DHydroxyapatite Ceramic Scaffold” [14].

### 3.3. Analysis of Institution Collaboration

To identify institutions with significant influence, we conducted an analysis of institutional co-occurrence based on relevant literature (Figure 6). Figure 6A displays the main institutional collaboration network of bone defects using bone xenografts, resulting in 285 nodes, 436 connections, and a density value of 0.0109. Figure 6B presents the publication volume and centrality of the top 10 institutions in terms of publication output. The institution with the strongest contribution and influence is the University of Zurich, with 29 publications and a centrality of 0.3, which indicates significant centrality. Other institutions that have made significant contributions include the University of Milan, the University of Sao Paulo, etc. In 2023, there was a collaboration between Peking University and China-Japan Friendship Hospital that resulted in the publication of a paper titled “Safety and Efficacy of Midface Augmentation Using Bio-Oss Bone Powder and Bio-Gide Collagen Membrane in Asians” [15], while institutions in Spain collaborated to publish “Physico-chemical and biological characterization of a new bovine bone mineral matrix available for human usage” [16].

### 3.4. Analysis of Author Collaboration

To identify highly influential authors in bone xenograft, we conducted a co-authorship analysis (Figure 7). We generated a co-citation network map comprising 476 nodes and 3210 connections (Figure 7A), with a density value of 0.0284. Figure 7B shows the citation counts and centrality of the top 10 most cited authors. Among them, Araujo MG was the most cited author in the past decade, with 163 citations and a centrality of 0.08. His most cited publication was “Dimensional ridge alterations following tooth extraction: An experimental study in the dog”, which explored the histological changes in alveolar bone absorption and remodeling after tooth extraction [17]. Froum SJ had the highest centrality with 46 citations and a centrality of 0.09. His most cited publication was “Vertical distance from the crest of bone to the height of the interproximal papilla between adjacent implants”, which found that the average height of the interproximal papilla between adjacent implants was 3.4 mm, ranging from 1 mm to 7 mm [18].

### 3.5. Analysis of Reference Burstiness and Clusters

We conducted burstiness and cluster analyses of our research data to identify highly central literature and provide a basis for analyzing research hotspots and trends (Figure 8 and Figure 9). Figure 8 shows the distribution of reference co-citations, with a total of 521 nodes and 2084 connections and a density value of 0.154. The top-ranked literature was a 2014 publication by Avila-Ortiz G et al. titled “Effect of alveolar ridge preservation after tooth extraction: a systematic review and meta-analysis” [19], with a centrality of 0.20. This study evaluated the impact of filling tooth extraction sockets with bone graft materials to prevent post-extraction alveolar ridge volume loss and found that alveolar ridge preservation (ARP) can effectively limit physiological atrophy compared to simple extraction [19].

The second-ranked literature, with a centrality of 0.16, was a 2016 publication by Vittorio Favero et al. titled “Sinus floor elevation outcomes following perforation of the Schneiderian membrane. An experimental study in sheep” [20]. This study assessed the effect of covering collagen membranes on bone formation after sinus membrane perforation in sheep and found that using collagen membranes on relatively small sinus mucosal perforations may result in greater new bone formation [20]. The third-ranked literature, with a centrality of 0.14, was a 2017 publication by Tim Fienitz et al. titled “Histological and radiological evaluation of sintered and non-sintered deproteinized bovine bone substitute materials in sinus elevation procedures. A prospective, randomized-controlled, clinical multicenter study” [21]. This study investigated the effectiveness of sintered and non-sintered bovine bone substitutes in maxillary sinus floor elevation (MSFE) and found that both groups had comparable new bone formation and volume stability [21]. We also conducted a clustering analysis of co-cited literature to identify the research foundation and focus areas. As shown in Figure 8, there are 10 main clusters, with earlier research hotspots on stromal cells, platelet-rich fibrin (PRF), socket preservation, and MSFE on the left. Recent hotspots appear on the right, including decellularization and multinucleated giant cells (MGCs), among others.

This study analyzed the top 10 most prominent publications in the past decade, with a focus on the evaluation of the effectiveness of ARP after tooth extraction [19,22,23,24,25] (Figure 9).

**Figure 9 bioengineering-10-00929-f009:**
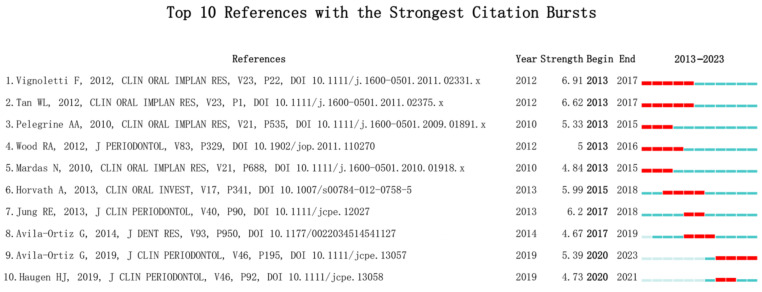
Top 10 References with the Strongest Citation Bursts [19,22,23,24,25,26,27,28,29,30].

### 3.6. Analysis of Keyword Burstiness and Clusters

We conducted a co-occurrence analysis on the acquired data to identify hot topics and urgent issues (Figure 10). To further obtain recent hot keywords, we applied the burst detection algorithm based on co-occurrence. A total of 298 nodes and 2733 connections were obtained, with a density value of 0.0618. The top three keywords in terms of frequency are “bone regeneration” (125 times), “tooth extraction” (116 times), and “augmentation” (112 times). The top two keywords in terms of centrality are “dental implant” (0.07) and “mesenchymal stem cell” (0.06).

We analyzed the top ten keywords with the highest burstiness level and found that three emerging keywords were “platelet rich fibrin”, “ridge augmentation”, and “management”. Among them, the most significant keyword was “platelet rich fibrin” (Figure 11).

Cluster analysis was performed on keywords, as shown in Figure 12, resulting in five main clusters: “tissue engineering”, “sinus floor elevation”, “dental implants”, “tooth extraction”, and “bone substitutes”. These clusters are primarily focused on dentistry.

## 4. Discussion

### 4.1. Global Research Status of Bone Xenograft

Bone grafts are defined as living tissues that promote bone healing and can be transplanted alone or in combination with other materials into bone defects [31]. The number of bone defect surgeries and related research has been increasing year by year [32,33]. Animal-derived bone grafts, widely used in clinical practice [34], have maintained stable development in the past decade (Figure 3), mainly focused on the field of dentistry (Figure 10). Among the 63 countries participating in the research, developed countries such as the United States, Germany, and Italy have a leading position in terms of publication volume and impact (Figure 4 and Figure 5), which is related to their early start of dentistry and a large proportion of medical investment. Notably, despite having fewer publications, France still has significant centrality (Figure 5), thanks to two high-quality papers that explored the effects of bone xenograft on periodontal ligament stem cells and MGCs, contributing to the study of the regenerative properties and biocompatibility of bone xenograft.

With economic development and improved medical level, developing countries like China and Brazil account for a relatively large proportion of research (Figure 4 and Figure 5). Professor Araujo MG from Brazil has made outstanding contributions (Figure 7), with his multiple studies on socket healing and ridge augmentation being widely cited. It is worth noting that the remaining five authors in the top 10 list had high centrality and citation counts, indicating their substantial contribution to the field of bone xenograft. Although China has not yet formed an institution or author with marking centrality (Figure 6), several Chinese research institutions recently cooperated in studying the safety and effectiveness of Bio-oss and Bio-guide for mid-facial augmentation in plastic surgery, promoting the research and application of xenogeneic bone graft materials in plastic surgery. While cooperation between countries is relatively close, it is mainly concentrated among countries with greater influence. However, the cooperation between countries with lower rankings remains relatively loose, and the development between countries is imbalanced (Figure 4). Therefore, for countries and regions with less influence, it is necessary to continue to increase financial investment, enhance institutional and international cooperation, and promote the research and development of bone xenografts.

### 4.2. Analysis of Hotspots and Trends in Bone Xenograft

The analysis of references reveals that bone xenograft has been a long-standing research hotspot in bone-augmentation technologies such as ARP, ridge augmentation, and MSFE. Come beam computed tomography (CBCT) has emerged as a popular non-invasive three-dimensional (3D) measurement method in bone xenograft research due to its convenience, non-invasiveness, and multi-dimensional advantages over clinical exploration and histological evaluations (Figure 8). Early studies indicate significant absorption of alveolar bone occurs within 3 to 6 months after tooth extraction [26]. To determine the effectiveness of bone-augmentation procedures and ensure implant stability, numerous researchers have studied the impact of various bone-augmentation techniques. CBCT enables researchers to quantitatively and intuitively measure and track changes in the contour of the alveolar ridge following tooth extraction and changes in bone elevation after MSFE [24,35], confirming the effectiveness of bone xenograft in various bone-augmentation techniques [36,37,38,39,40,41,42].

Ideal bone graft materials for bone-augmentation techniques should possess properties such as biocompatibility, good mechanical performance, osteoconductive, osteoinductive capabilities, and appropriate degradation rates [30]. Bone xenografts from various sources can effectively promote the healing of bone defects by embedding newly formed bone, exhibit low degradation rates, and provide excellent mechanical support to maintain bone repair space and implant stability [43,44,45,46,47]. However, the osteoinductivity of bone xenografts has not yet reached the ideal standards achieved by autologous or allogeneic bone materials [42,47,48,49,50,51]. Heat treatment or sintering is a commonly used physical purification method for bone xenografts. Research findings reveal that xenograft bone materials purified by high-temperature sintering could cause more numerous MGCs to aggregate under high temperatures, leading to pro- or anti-inflammatory effects that may affect tissue regeneration [13,52,53]. MGCs continue to be one of the major areas of research focus, and their roles and impacts on tissue regeneration and material degradation require further study (Figure 8). In this context, further investigation and improvement are necessary to enhance the tissue reactivity and osteoinductivity of bone xenografts in the future. In addition to analyzing references, we conducted reference co-citation analysis and found that decellularization technology is an emerging research hotspot in recent years (Figure 8). Decellularization is a process that removes cells and their related components from the extracellular matrix (ECM) through physical, chemical, or enzymatic treatments, especially DNA and RNA, to produce natural matrices with intact mechanical properties [54]. This technology is also one of the pathways used for the purification of bone xenografts [55]. Porcine bone tissue has similar bone microstructures to humans. However, the presence of α1, 3 Gal epitopes can cause hyperacute rejection in humans [56,57]. Therefore, researchers have employed decellularization technology to purify porcine bone xenografts [58]. Compared to deproteinized bovine bone grafts prepared via high-temperature sintering (300–1300 °C), the decellularized porcine bone ECM retains various proteins that support intracellular and extracellular signaling pathways, such as Chondroadherin, Lumican, and Biglycan. Additionally, it preserves the natural microstructure as much as possible, showing better new bone regeneration ability in preclinical studies [59]. Researchers have used 3D printing technology to prepare xenogeneic composite scaffolds by combining decellularized porcine bone with polycaprolactone, resulting in scaffolds with lower immunogenicity, better osteogenic performance, and higher degradation rates [56]. Considering these results, decellularized xenogeneic bone grafts hold immense potential for repairing and regenerating bone defects. However, various decellularization methods may still damage the ECM or not eliminate immune risks due to residual cell components [60,61]. Future clinical studies and applications of decellularization in bone defect repair must focus on providing long-term experimental data and conducting more clinical trials with the development of molecular biology and cell biology, tissue engineering has gradually delved into the study of bone xenografts, which is one of the long-term research directions (Figure 12). Mesenchymal stem cells (MSCs) have gained extensive attention due to their ability for self-renewal and multipotent differentiation [62]. Several preclinical or clinical studies have shown that MSCs from different sources, such as adipose tissue and bone marrow, significantly promote bone regeneration of xenografts [63,64,65]. However, some scholars believe that cell therapy still faces issues such as high cost, inconvenient regulation, and long treatment times. Further optimization is needed in aspects such as cell selection, delivery, vitality, and phenotypic stability before it can be widely applied in clinics [66,67]. Therefore, the concept of a cell-free strategy has been receiving increasing attention from scholars. The cell-free bone biomimetic scaffold aims to interact with surrounding cells and tissues through the scaffold and its loaded growth factors or other bioactive substances to promote vascularization and osteogenesis, thereby changing the traditional recovery process of diseases or injuries [68].

In the past 10 years, researchers have attempted various bioactive substances to enhance the osteoinductive and osteoconductive properties of xenografts, including bone morphogenetic protein 2, parathyroid hormone, 4-hexylresorcinol, and titanium particles, all of which have achieved certain results [64,69,70,71,72,73,74,75]. Through the analysis of keyword burstiness, we found that platelet-rich fibrin (PRF) has become one of the emerging research hotspots of bone xenograft in the past three years (Figure 11). PRF is a second-generation autologous platelet concentrate named PRF in 2005 and is obtained by the centrifugal processing of blood. It is rich in platelets, white blood cells, mononuclear cells, and various cytokines, with advantages such as simple preparation, no immunogenicity, and easy clinical application [76,77]. In vitro, studies have shown that PRF exhibits properties such as promoting angiogenesis and has achieved good results when used in conjunction with xenografts in studies on ARP, guided bone regeneration, periodontal regeneration, MSFE, and accelerated orthodontic tooth movement [15,46,78,79,80,81]. Studies have found that different centrifugal processes may alter the 3D network structure of PRF, which may potentially affect the efficacy of related bone-augmentation techniques [72,73,74,75,76,77,78,79,80,81,82,83,84]. Based on the above background, PRF has great research potential in enhancing the performance of bone xenografts and is one of the future research hotspots.

### 4.3. Advantages and Limitations

Bone xenografts are one of the major bone substitute materials used clinically. This study focuses on the analysis of the current research status of bone xenografts, and the research results can provide data support and new ideas for the improvement of new bone grafts. However, there are still some limitations in this study. Firstly, the data sources only include the literature published in the WoSCC database, which may miss some relevant studies from other databases, affecting the comprehensiveness of this study’s analysis. Secondly, the citation count of literature may increase over time, which may result in relatively lower citation counts for some recent important literature and not being prominently reflected in the analysis results.

## 5. Conclusions

In conclusion, between 2013 and 2023, the research scale of bone xenograft has experienced a steadily increased. The research hotspot and directions are mainly focused on the dental field, encompassing bone-augmentation techniques related to implants and bone tissue engineering. The application of CBCT has significantly supported the development of this field, eliciting anticipation for the application of new devices and technologies in future research.

The development of various decellularization techniques is one of the current research directions. However, promoting the clinical application of these technologies requires further studies due to the lack of relevant clinical research.

PRF has been the focus point field of bone xenograft in recent years. In addition to various clinical studies, significant research space is available for adjusting the 3D network structure and exploring its effects.

## Figures and Tables

**Figure 1 bioengineering-10-00929-f001:**
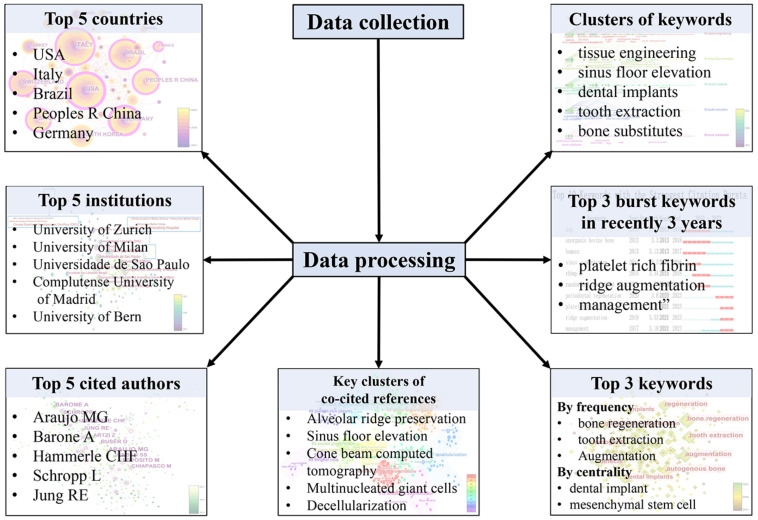
Flow diagram and results of bibliometric analysis.

**Figure 2 bioengineering-10-00929-f002:**
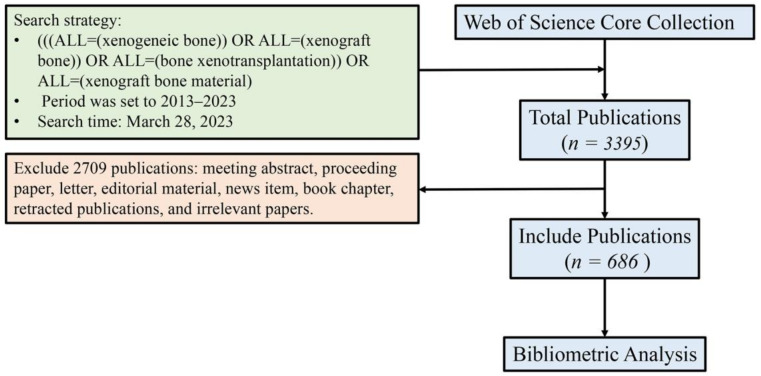
Process of literature search and filtration.

**Figure 3 bioengineering-10-00929-f003:**
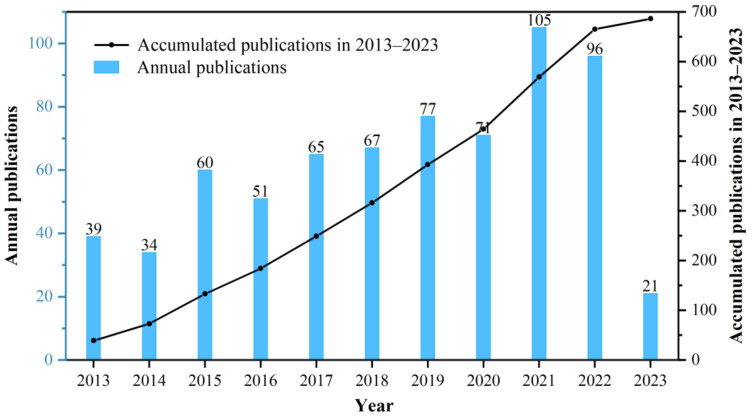
The annual trends of publications during 2012–2023.

**Figure 4 bioengineering-10-00929-f004:**
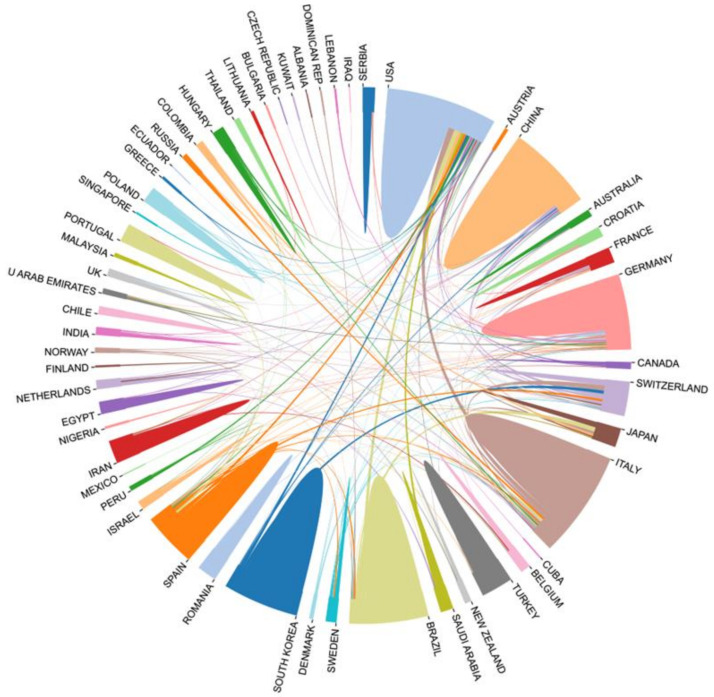
National cooperation network diagram.

**Figure 5 bioengineering-10-00929-f005:**
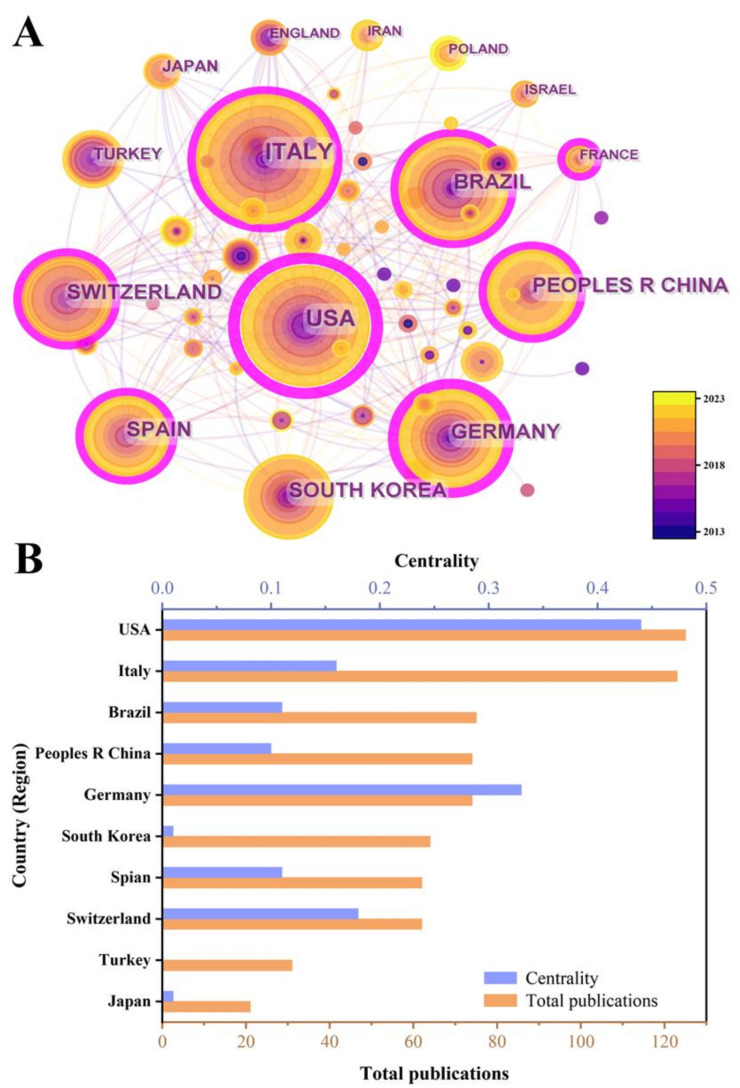
The publication number of the top 10 countries. (**A**) Knowledge map visualized by CiteSpace. (**B**) Bar chart.

**Figure 6 bioengineering-10-00929-f006:**
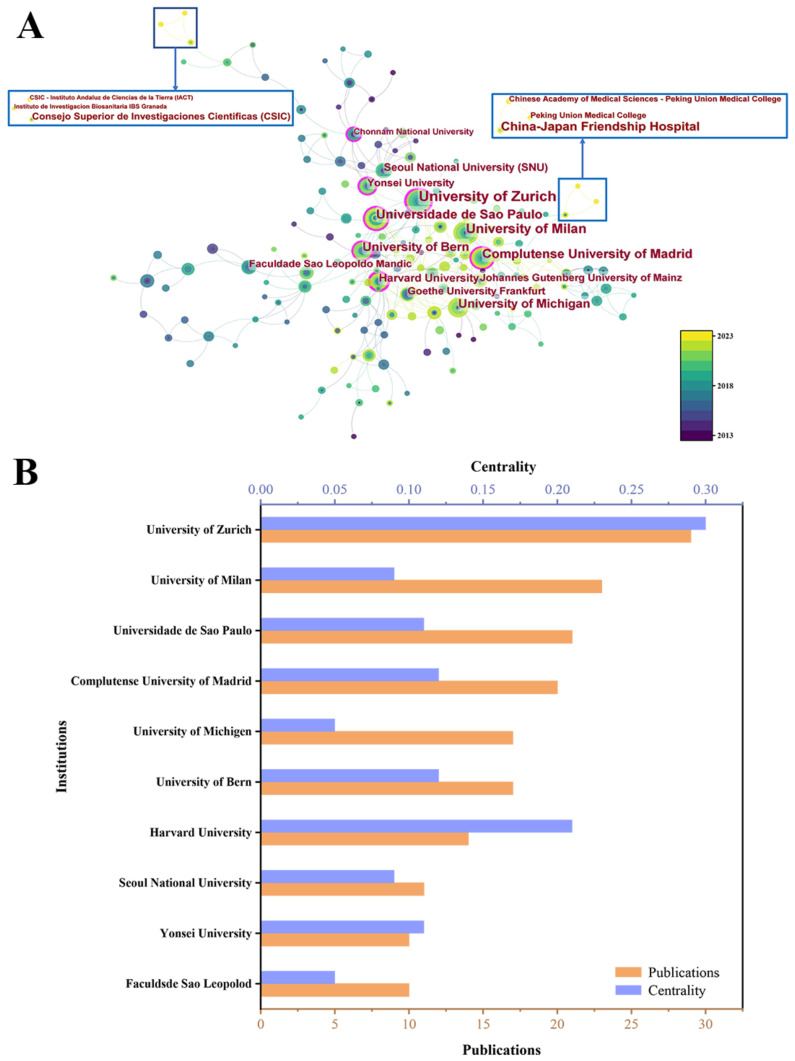
(**A**) Institution cooperation network diagram. (**B**) The publication number of the top 10 institutions.

**Figure 7 bioengineering-10-00929-f007:**
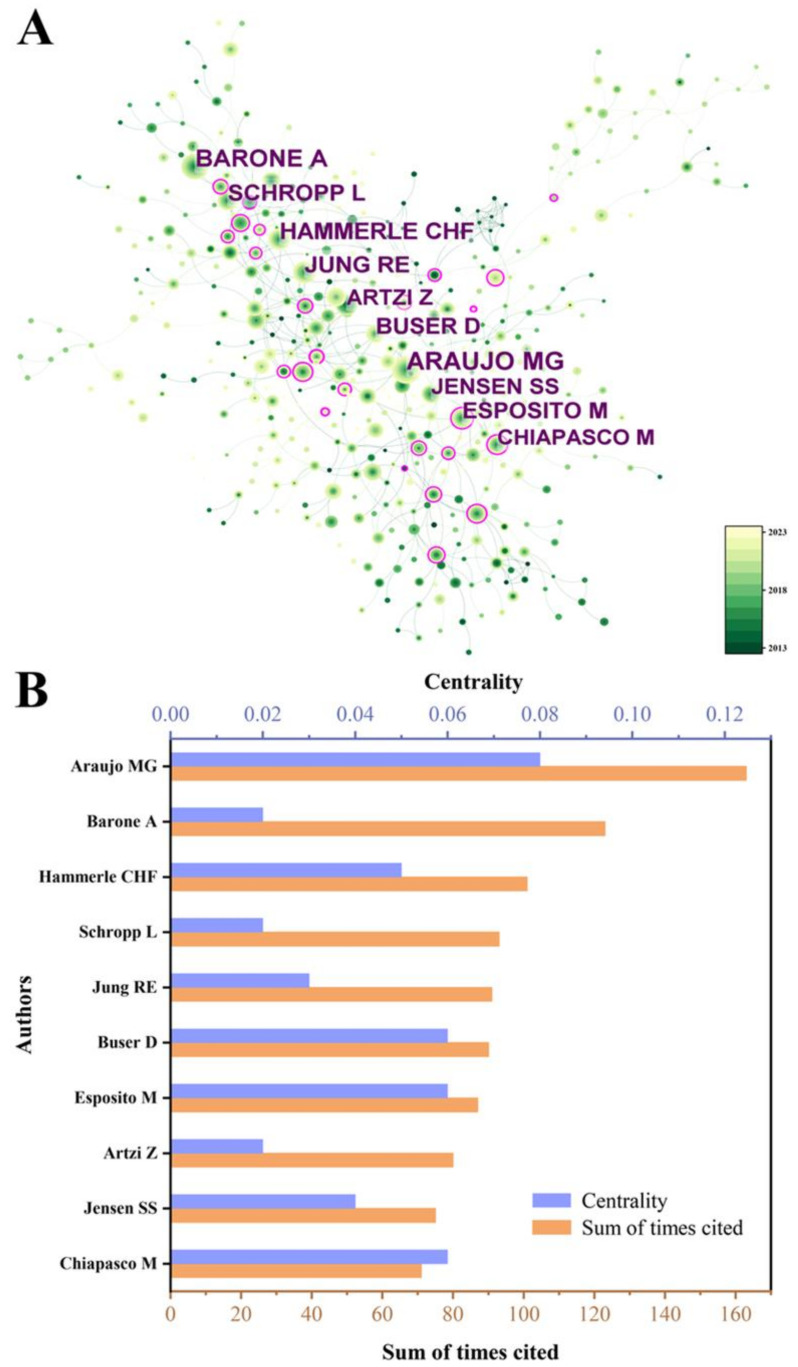
(**A**) The author co-cited distribution map. (**B**) The cited times of the top 10 authors.

**Figure 8 bioengineering-10-00929-f008:**
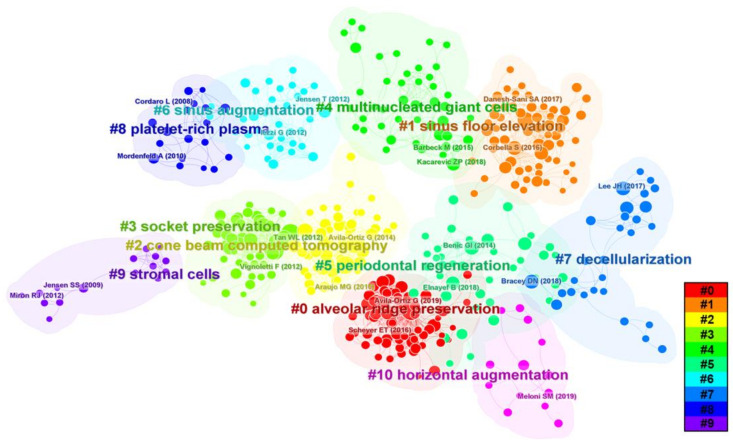
References co-cited distribution map.

**Figure 10 bioengineering-10-00929-f010:**
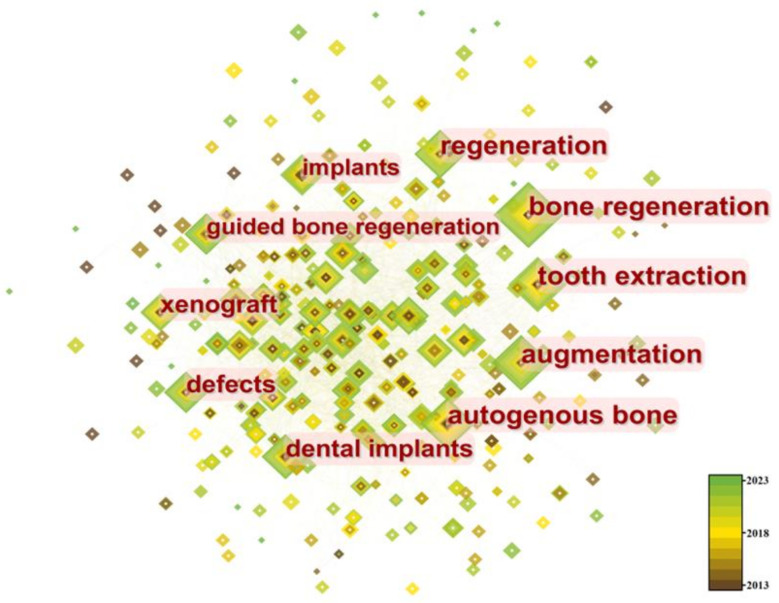
The co-occurrence network of keywords.

**Figure 11 bioengineering-10-00929-f011:**
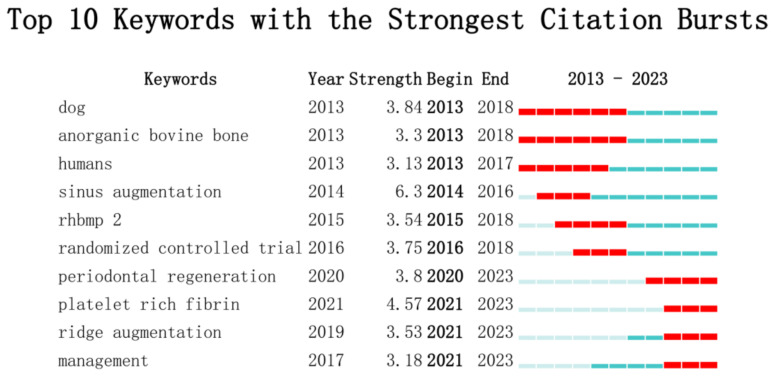
Top 10 Keywords with the Strongest Citation Bursts.

**Figure 12 bioengineering-10-00929-f012:**
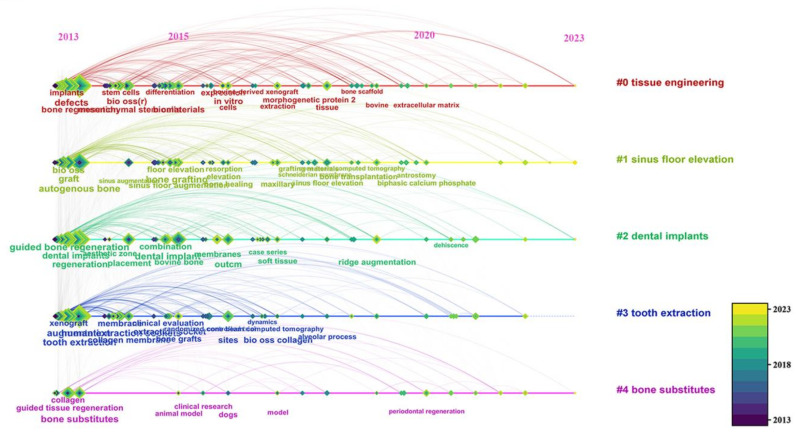
Time distribution of keywords in different clusters.

## Data Availability

The data presented in this study are available on request from the corresponding author.

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
