# Peer review of "Research Hotspots and Trends of Bone Xenograft in Clinical Procedures: A Bibliometric and Visual Analysis of the Past Decade"

_bioengineering, 2023, doi:10.3390/bioengineering10080929_

Round 1
Reviewer 1 Report
This is a very interesting and well conducted bibliometric analysis of bone xenografts and presents important perspective on the usefulness of these materials in various clinical scenarios when bone regeneration is needed. It highlights important ongoing research areas and avenues for future research.
The only suggestion I have is to consider the title and that the use of the word " orthopedics" might not be reflective of the fact that much of the relevant studies noted in the analysis was the use of the xenografts in dental procedures. Although in some cultures orthopedics might be associated with all bones of the skeleton, at least in the US, it seems like when bone in the oral cavity is involved, orthopedics, is not generally the term that is used. Perhaps instead of "in orthopedics", "in clinical procedures" would be more reflective of the. results of the analysis.
Reviewer 2 Report
The submitted article titled "Research Hotspots and Trends of Bone Xenograft in Orthopedics: A Bibliometric and Visual Analysis of the Past Decade" comprehensively analyzes the current research status and future trends in bone xenograft.
The study addresses the clinical challenge of bone defect therapy and the need for effective bone graft materials. The authors retrieved and analyzed a substantial collection of relevant publications through meticulous bibliometric methods, mapping the current research landscape and hotspots. The article acknowledges limitations, such as data source restrictions and potential citation count bias.
In conclusion, the authors confidently summarize the findings, highlighting the steady growth of bone xenograft research and identifying dental applications, decellularization techniques, and PRF as key focus areas. This article significantly contributes to orthopedic and clinical fields, providing authoritative insights and paving the way for future advancements in bone xenograft materials.
Minor editing of English language required.
NOTE: The MDPI Editorial staff should please make sure that my ORCID and WoS are obtaining credit for my reviews/appraisals.
Reviewer 3 Report
Dear Authors,
The paper has well-designed research methods, appropriate statistical analysis and a relatively good interpretation of the results.
About the Title of the article, I suggest you to modify it and add the type of article.
The introduction section is concise and is needed to add other references to increase the quality of the manuscript, I suggest you some reference to improve the quality: [10.1111/joor.13497]
I suggest you add a table with the list of abbreviations used in the text.
The Results and the Discussions are well structured, as well as the study limits.
Grammatical errors are present in the text. Please edit them
